# Prevalence and predictive risk factors of hypertension in patients hospitalized in Kamenge Military hospital and Kamenge University teaching hospital in 2019: A fixed effect modelling study in Burundi

Arnaud Iradukunda[1,2,3,4]*, Emmanuel Nene Odjidja[3,5], Stephane Karl Ndayishima[6], Egide Ngendakumana[7], Gabin Pacifique Ndayishimiye[2], Darlene Sinarinzi[2,8], Cheilla Izere[4,9], Nestor Ntakaburimvo[2,4], Arlene Akimana[10,11]

1 Department of Medicine, University of Burundi, Bujumbura, Burundi, 2 Department of Statistics, Lake Tanganyika University, Mutanga, Burundi, 3 Royal Society of Tropical Medicine and Hygiene, London, United Kingdom, 4 Department of Research and Innovation, ARNECH Research and Consulting Office, Bujumbura, Burundi, 5 Department of Medicine, School of Clinical Sciences, Monash University, Clayton, VIC, Australia, 6 Departemend de Medecine, université Paris-Est-Créteil-Val-de-Marne, Créteil, France, 7 Department of Applied Econometrics, University of Lille, Villeneuve-d'Ascq, France, 8 Departemente de Suivie-Evaluation des projects, Instututs Sciences Campus du Centre, Ouagadougou, Burkina-Fasso, 9 Department of Computer Mathematics, Clermont Auvergne University, Clermont-Ferrand, France, 10 Institut Universitaire des Sciences de la Santé et du Développement, Bujumbura, Burundi, 11 Department of Public Health, Aix-Marseille University, Marseille, France

* arnaudiradukunda5@gmail.com

**Data Availability Statement:** All relevant data are within the manuscript and its Supporting Information files

## Abstract

### Introduction

Hypertension is a major threat to public health globally. Especially in sub-Saharan African countries, this coexists with high burden of other infectious diseases, creating a complex public health situation which is difficult to address. Tackling this will require targeted public health intervention based on evidence that well defines the at risk population. In this study, using retrospective data from two referral hospitals in Burundi, we model the risk factors of hypertension in Burundi.

### Materials and methods

Retrospective data of a sample of 353 randomly selected from a population of 4,380 patients admitted in 2019 in two referral hospitals in Burundi: Military and University teaching hospital of Kamenge. The predictive risk factors were carried out by fixed effect logistic regression. Model performance was assessed with Area under Curve (AUC) method. Model was internally validated using bootstrapping method with 2000 replications. Both data processing and data analysis were done using R software.

### Results

Overall, 16.7% of the patients were found to be hypertensive. This study didn't showed any significant difference of hypertension's prevalences among women (16%) and men

**Funding:** The authors received no specific funding for this work.

**Competing interests:** The authors have declared that no competing interests exist.

**Abbreviations:** AIC, Akaike Information Criterion; AOR, Adjusted Odds Ratio; AUC, Area Under Curve; BMI, Body Mass Index; BP, Blood Pressure; CI, Confidence Interval; CKF, Chronic Kidney failure; Df, Degrees of freedom; CS, Clinical symptoms; FHH, Family History with Hypertension; HTN, Hypertension; H-, Normotensive people; H+, Hypertensive people; OR, Odds Ratio; ROC, Receiver Operating Characteristic; SSA, Sub-Saharan Africa; WHO, World Health Organisation.

(17.7%). After adjustment of the model for cofounding covariates, associated risk factors found were advanced age (40–59 years) and above 60 years, high education level, chronic kidney failure, high body mass index, familial history of hypertension. In absence of these highlighted risk factors, the risk of hypertension occurrence was about 2 per 1000 persons. This probability is more than 90% in patients with more than three risk factors.

## Conclusion

The relatively high prevalence and associated risk factors of hypertension in Burundi raises a call for concern especially in this context where there exist an equally high burden of infectious diseases, other chronic diseases including chronic malnutrition. Targeting interventions based on these identified risk factors will allow judicious channel of resources and effective public health planning.

## Introduction

Hypertension corresponds to a permanently raised blood pressure in arteries and arterioles [1]. It is defined as a systolic blood pressure equal or above 140 mmHg and /or a diastolic blood pressure above 90 mmHg [1, 2]. Hypertension is a threat to global public health [3] as it tires vessels, the heart and causes damage to artery walls [4, 5]. It is a major risk factor for cardiovascular diseases [6] with high morbidity and mortality rate [6]. If not identified and treated early, arterial hypertension may result in serious complications including strokes, coronary artery, kidney and hypertensive heart diseases [1, 7]. These complications are among the leading causes of mortality in the world. Approximately, cardiovascular diseases account for 17.8 million death in 2017 [8, 9], nearly 1/3 of total. More than three quarters were in low and middle-income, countries (LMICs) [9]. Hypertension complications, cardiovascular diseases account 9.4 million (52.8%) every year [10]. Hypertension is responsible for 45% deaths due to heart disease and 51% stroke related deaths [11, 12]. Premature death and health care expenditure for treatments due the hypertension puts an economic toll on families and pushes many into poverty [13, 14]. At the macro level, these high expenses and human losses significantly impacts on economic growth and reduces productivity [15, 16].

The prevalence of hypertension in adults was 40% only in 2015, with an estimated 1.13 billion people living with different forms of hypertension [17, 18]. Data from the World Health Organisation Global Health Observatory Repository [19] found the highest prevalence of hypertension in the Africa region (46%) followed by the America (35%) and other regions, majority of whom, were undiagnosed and untreated [20, 21]. In sub-Saharan Africa (SSA), as other settings, hypertension has been associated with lifestyles, diets, physical inactivity urbanization and socio economic status [22]. More than 125 million people with hypertension are expected by the year 2025 in SSA alone [23]. By year 2030, hypertension and other non-communicable diseases are projected to surpass communicable diseases as the top of mortality causes on the continent [24]. From 2011 to 2025, the cumulative lost output with non-communicable diseases is projected to be US$7.28 trillion in LMICs which is approximately a loss of US$500 billion per year [25]. Cardiovascular diseases including hypertension account for nearly half this cost [26]. Despite this, SSA faces a major problem of early screening, timely treatment and control of hypertension [27, 28]. Yet, studies to understand the epidemiology and associated risk factors of hypertension in the context of Burundi are lacking, prompting the conduct of this study. Therefore, in this study, we determined the overall prevalence of

hypertension. We also evaluated its predictive risk factors and its occurrence probability based on risk factors. Knowing these factors could support effective public health planning and facilitate policy makers to formulate plausible policies towards the fight against hypertension and its complications.

## Materials and methods

### Ethics statement

This investigation is in accordance with Helsinki's Declaration and approved by the institutional Research Ethics committees. Written permissions were acquired from ethical committee of the University teaching hospital of Kamenge and Kamenge military hospital to use retrospective data for this study. In effort to secure identity of patients, patient's information were anonymised and replaced by a unique codes. Data are used for the unique objective of this study and will be destroyed after paper publication.

### Study description

We employed a cross-sectional study. The study was conducted in two tertiary reference hospitals of Burundi: University teaching hospital of Kamenge and Kamenge Military hospital in different departments. Patients were hospitalized for different medicals diseases including infectious diseases, cardiovascular diseases and others. Based on socio-demographic and clinical characteristics associated with hypertension, the study used data from internal medicine services with or not haemodialysis sessions (for patients with kidney failure) where majority of patients with hypertension and others cardiovascular diseases are admitted Data were collected among patients hospitalized from January 2019 to December 2019 in these services. These hospital were firstly chosen because they are ranked as tertiary national reference and their receive patients from all over the country. Secondly, one is private hospital and another a public hospital.

### Sampling methods

This study targeted a population of 4380 patients hospitalized in internal medicine services and intensive care unit (with chronic kidney failure) during the whole study period in the both hospitals. Majority of these patients are hospitalized there because this areas are located together with haemodialysis unit. Among them, a sample size of 353 patients have been selected. The inclusion probability were the same for patients admitted in the same service of the selected hospital and was calculated as ratio of admitted patients in the service over all admissions of all services. Patients were randomly selected in the service using their specific identifiers of admission. The minimal sample size was calculated based on the quintile of the normal distribution with at 95% of confidence (1.96), the population size, the prevalence of hypertension and the acceptable margin (5%) [28]. As the prevalence of hypertension is unknown in Burundi, a value of $p = 0.5$ was selected. According to these parameters, the minimal size of the sample was 353 patients. Basically, a respondent was selected if he had of the following criteria: Admitted in targeted service in the period of 2019, measured diastolic and systolic blood pressure three times.

Normally, after resting in a quiet environment for 5–10 minutes, the patient took a sitting position with his legs naturally flat and his right hand placed at heart height. An Omron HEM-705P arm-type electronic sphygmomanometer were used to measure the systolic blood pressure, diastolic blood pressure and resting heart rate of the brachial artery of the right upper

arm at least twice, with an interval of no less than two minutes. The average of the two readings was taken.

## Data collection

In total, a population of 4380 patients stratified in 2 groups in both hospitals were targeted. A random sampling method have been used with proportional allocation to the admitted patients by service and by hospital. Data were collected using a standard-structured questionnaires based on hospital record. We collected socio-demographics, anthropometrics and clinical characteristics as summarized in **Table 1**. All patients with hypertension appeared before chronic kidney failure were excluded in the study. In this context, chronic kidney failure is predictor of hypertension.

## Outcome and independent variables

In this study, hypertension was considered as outcome variable and was defined as systolic blood pressure ≥ 140 mmHg and/or diastolic blood pressure ≥ 90 mmHg obtained after three

**Table 1. Descriptive analysis of patients' characteristics.**

| Individual characteristics | Category | N | H⁻ | H⁺ | P+ | 95% CI of P+ |
|---|---|---|---|---|---|---|
| Sex | Men | 206 | 173 | 33 | 16.0 | [11.3–21.8] |
|  | Women | 147 | 121 | 26 | 17.7 | [11.9–24.8] |
| Age group | 15–39 years | 138 | 135 | 3 | 2.2 | [0.5–6.2] |
|  | 40–59 years | 131 | 103 | 28 | 21.4 | [14.7–29.4] |
|  | ≥60 years | 84 | 56 | 28 | 33.3 | [23.4–44.5] |
| Marital status | Married | 89 | 81 | 8 | 9.0 | [39.0–16.9] |
|  | Unmarried | 264 | 213 | 51 | 19.3 | [14.7–24.6] |
| Educational level | Primary or less | 137 | 123 | 14 | 10.2 | [5.7–16.6] |
|  | Secondary | 165 | 135 | 30 | 18.2 | [12.6–24.9] |
|  | University | 51 | 36 | 15 | 29.4 | [17.5–43.8] |
| Alcohol | No | 173 | 145 | 28 | 16.2 | [11.1–22.5] |
|  | Yes | 180 | 149 | 31 | 17.2 | [12.0–23.6] |
| Smoking | No | 325 | 274 | 51 | 15.7 | [11.9–20.1] |
|  | Yes | 28 | 20 | 8 | 28.6 | [13.2–48.7] |
| Existing cardiovascular impairment | No | 28 | 24 | 4 | 14.3 | [4.0–32.7] |
|  | Yes | 325 | 270 | 55 | 16.9 | [13.0–21.5] |
| Family History with Hypertension | No | 289 | 262 | 27 | 9.3 | [6.3–13.3] |
|  | Yes | 64 | 32 | 32 | 50.0 | [37.2–62.7] |
| Diabetes | No | 273 | 247 | 26 | 9.5 | [6.3–13.6] |
|  | Yes | 80 | 47 | 33 | 41.3 | [30.4–52.8] |
| Chronic kidney failure | No | 169 | 164 | 5 | 3.0 | [1.0–6.7] |
|  | Yes | 184 | 130 | 54 | 29.3 | [22.9–36.5] |
| Body mass index | Underweight and normal | 270 | 236 | 34 | 12.6 | [8.9–17.3] |
|  | Overweight | 60 | 44 | 16 | 26.7 | [16.5–39.9] |
|  | Obesity | 23 | 14 | 9 | 39.1 | [20.5–61.2] |
| Overweight | No | 271 | 237 | 34 | 12.5 | [8.9–17.1] |
|  | Yes | 82 | 57 | 25 | 30.5 | [20.8–41.7] |
| **Total** |  | **353** | **294** | **59** | **16.7** | **[15.6–25.1]** |

N: Category's total, H⁻: Normotensive people, H⁺: Hypertensive people, P⁺: Hypertensive people proportion

successive measurements or patients on hypertension's treatment. Pre-hypertension was defined as BP levels of 120 to 139/80 to 89 mm Hg [29]. Body mass index (BMI) was calculated as the weight in kilograms to square of height in meters and was categorized in into; underweight (BMI<18.5 kg/m$^2$), normal (BMI: 18.5–24.9 kg/m$^2$), overweight (BMI: 25.0–29.9 kg/m$^2$) and obesity (BMI $\geq$ 30 kg/m$^2$). During data analysis, BMI was categorized in three clusters (Under and normal weight, overweight, obesity). Others independents variables were classified as age group (in years) (15–39 years, 40–60 years; $\geq$60) sex (Men, Women), residence (Urban, Rural), educational level (Primary or less, Secondary, University), alcohol consumption (Yes/No), smoking (Yes/No), diabetes (Yes/No), existing cardiovascular impairment (Yes/No), familial history of hypertension (Yes/No) and chronic kidney failure (Yes/No). As chronic kidney failure can be predictor or consequence of hypertension, only patients with history of kidney failure before hypertension were included in the study.

## Statistical analysis

Data analysis were undertaken in different steps: descriptive analysis, binary logistic modelling with fixed effects, power predictive evaluation of final (saturated) model and probabilities prediction. Hypertension associated risk factors were assessed using univariate and multivariate logistic regressions. We calculated the odds ratios (ORs) at 95% confidence level for each covariate to identify predictors of hypertension. Significant variables on 15% threshold were used in multivariable logistic modelling to determine a combined effect on the outcome. The likelihood ration test, the score test and the Wald test were used to determine significance of independent variables on the outcome [30]. A threshold of 5% have been used. To select the best model for this study, we use backward method with parsimony principle [31]. The Akaike Information Criterion (AIC) based on adjustment were used [32]. The best model is one with low AIC value.

The relevance of the final model to make prediction was assessed by Pearson residuals test. The Receiving Operating characteristics (ROC) and Area under Curve (AUC) were respectively used to compare and evaluate performance and predictive power of the model. Furthermore, the ROC was used to determine the discriminatory performance of the model, determining the false positive and false negative rates. The Mann Whitney statistics method showed that the two distributions were offset: normotensive people had an average higher scores than hypertensive people. Each individual's score was ranked in ascending order. Thus, the AUC which determined the number of observations accurately predicted was calculated based on the number of normotensive patients, the number of hypertensive patients and the ranks of normotensive patients. If AUC = 0.5 no discrimination, 0.7$\leq$AUC$\leq$0.8 acceptable discrimination, 0.8$\leq$AUC<0.9 excellent discrimination, AUC$\geq$0.9 exceptional discrimination. To calculate the error of predictions, the cross validation with K -Fold method have been used. We used 10-folds. In validating internal performance of the model, a bootstrapping procedure with 2000 replications was done. Statistical significance was considered when p-value $\leq$ 0.05.

Influents points of the model were assessed using Hoaglin and Welsh criterion, methods based on model's parameters and the sample size. The adjustment used was based on the residuals test. The Kolmogorov test was used to test the model adequation. For these test, a threshold of 5% was used. Then, the foreign, rpart and forest model packages were used to carry out results in this study [33]. Data processing and data analysis were done using R.3.5.0 software.

## Results

In this study, the overall prevalence of hypertension was 16.7% (Table 1). Comparatively, men's and in women's prevalence were respectively 16.0% and 17.7% with insignificant

difference ($X^2$ = 0.0725, df = 1, p = 0.788). This prevalence was 2 times higher in overweight patients than normal and underweight patients and 3 times higher in diabetic patients than no diabetics patients. The pre hypertension was observed in 73.4% of patients. The high proportions above the overall prevalence were observed in people with existing cardiovascular impairment, married, patients aged between 40–60 years or 60 years and above and over, patients with chronic kidney failure, men patients, smokers patients, obese patients and patients with secondary or university level (**Table 1**). The table below showed descriptive analyses of patients' characteristics.

## Logistic regression modelling of predictive risk factors of hypertension

The Table 2 showed the univariate logistic regressions where hypertension was modelled by each explanatory variable.

Eight variables were significantly associated with hypertension in univariate regression (**Table 2**). Those variables were Educational level{University (OR: 3.7, 95% CI: 1.6–8.4, p<0.002); Secondary (OR: 1.9, 95% CI: 1.1–3.9, p: 0.045), Diabetes (OR: 6.7, 95% CI: 3.37–

**Table 2. Univariates logistic regressions of hypertension's factors.**

| Variables | Category | OR | 95% CI | p |
|---|---|---|---|---|
| Educational | Primary or less | Reference | | |
| level | Secondary | 1.9 | [1.0–3.9] | 0.054 |
| | University | 3.7 | [1.6–8.4] | 0.002 |
| Smoking | No | Reference | | |
| | Yes | 2.1 | [0.9–05.0] | 0.085 |
| Diabetes | No | Reference | | |
| | Yes | 6.7 | [3.7–12.3] | <0.001 |
| Age group | 15–39 years | Reference | | |
| | 40–59 years | 12.2 | [4.2–52.1] | <0.001 |
| | ≥60 years | 22.5 | [17.6–96.8] | <0.001 |
| Currently working | No | Reference | | |
| | Yes | 0.9 | [0.6–1.7] | 0.944 |
| Residence | Rural | Reference | | |
| | Urban | 1.1 | [0.6–1.8] | 0.868 |
| Body mass index | Underweight and normal | Reference | | |
| | Overweight | 2.5 | [1.3–4.9] | 0.007 |
| | Obesity | 4.5 | [1.7–10.9] | 0.001 |
| Sex | Women | Reference | | |
| | Men | 0.8 | [0.6–1.9] | 0.451 |
| Marital status | Unmarried | Reference | | |
| | Married | 2.4 | [1.2–5.7] | 0.027 |
| Existing cardiovascular impairment | No | Reference | | |
| | Yes | 1.2 | [0.5–4.3] | 0.720 |
| Alcohol | No | Reference | | |
| | Yes | 1.1 | [0.6–1.9] | 0.794 |
| Chronic Kidney | No | Reference | | |
| failure | Yes | 13.6 | [5.8–39.9] | <0.001 |
| Familial History | No | Reference | | |
| of Hypertension | Yes | 9.7 | [5.2–18.4] | <0.001 |

OR: Odd ratio, CI: Confident interval, p: p-value.

12.3, p<0.001), Age {(40–59 years (OR: 12.2, 95% CI: 4.2–22.1, p<0.001); 60 and above (22.5, 95% CI: 7.6–26.8, <0.001)}, Body mass index {Overweight (OR: 2.5, 95% CI: 1.3, 4.9, p: 0.007); Obesity (OR: 4.5, 95% CI: 1.7–10.9, p: 0.001)}, Marital status (OR: 2.4, 95% CI: 1.2–5.7, p: 0.027), Chronic kidney failure (OR: 13.6, 95% CI: 5.8–19.9, p<0.001) and Family hypertension history (OR: 9.7, 95% CI: 5.2–18.4, p<0.001). All these variables were introduced in a multivariate analysis. In addition, other relevant variables were introduced in the multivariate model as well as sex, smoking, alcohol and existing cardiovascular impairment. The identification of variables based on AIC and backward selection methods showed six variables significantly associated with hypertension.

Then, the equation of saturated model derived from the above multivariate model include the Educational level, Body mass index, Chronic kidney failure and familial history of hypertension. After controlling the cofounding variables as indicated above, adjusted odds ratio and their lower and upper bound along with a corresponding p-value were derived. Table 4 detailed these results.

The Table 3 showed that advanced age and education level were the most associated risk factor of hypertension. Chronic kidney failure, overweight, obesity, familial hypertension history were significantly associated with hypertension. Patients with secondary level had near five times higher (AOR: 4.9, 95% CI: 2.1–12.4, p<0.001) the risk of hypertension occurrence than patients with primary or less educational level. That risk was more than 8 times higher (AOR: 8.6, 95% CI: 2.9–27.3, p<0.001) in patients with university level than patients with none or primary level. Even if smokers had more than two times high the risk (AOR: 2.9, 95% CI: 0.8–9.3, p: 0.073) to become hypertensive than non-smokers, tobacco consumption were not significantly associated with hypertension. Tobacco consumption remained in saturated model because it minimized the AIC criterion than the model with no tobacco use as explanatory variable. Patients aged between 40–59 years had more than 5 times (AOR: 5.9, 95% CI: 1.9–26.8, p: 0.007) the risk to become hypertensive than patients aged under 40 years. That risk was more than 12 times higher (AOR: 12.9, 95% CI: 3.4–65.8, p<0.001) in patients aged

**Table 3. Results of multivariate logistic regression.**

| Variables | Category | N | H⁻ | H⁺ | AOR | 95% CI | p |
|---|---|---|---|---|---|---|---|
| Education level | Primary or less | 137 | 123 | 14 | Reference | | |
| | Secondary | 165 | 130 | 35 | 4.6 | [1.9–11.5] | <0.001 |
| | University | 51 | 36 | 15 | 8.5 | [2.9–27.2] | <0.001 |
| Smoking | No | 325 | 274 | 51 | Reference | | |
| | Yes | 28 | 20 | 8 | 2.9 | [0.9–9.3] | 0.073 |
| Age group | 15–39 years | 138 | 135 | 3 | Reference | | |
| | 40–59 years | 131 | 103 | 28 | 5.9 | [1.8–26.8] | 0.007 |
| | ≥60 years | 84 | 56 | 28 | 12.9 | [3.4–65.8] | <0.001 |
| Body mass index | Underweight and normal | 270 | 236 | 34 | Reference | | |
| | Overweight | 60 | 44 | 16 | 2.5 | [1.1–5.7] | 0.037 |
| | Obesity | 23 | 14 | 9 | 3.7 | [1.2–11.5] | 0.023 |
| CKF | No | 169 | 164 | 5 | Reference | | |
| | Yes | 184 | 130 | 54 | 4.9 | [1.8–15.6] | 0.003 |
| FHH | No | 289 | 262 | 27 | Reference | | |
| | Yes | 64 | 32 | 32 | 2.9 | [1.3–6.7] | 0.014 |

H⁻: Normotensive people, H⁺: Hypertensive people, AOR: Adjusted Odd ratio, CI: Confidence interval, p: p-value, CKF: Chronic kidney failure, FHH: Familial history of hypertension.

**Table 4. Predicted probabilities.**

| Ind. | Education level | Smoking | Age group | BMI | CKF | FHH | p |
|------|-----------------|---------|-----------|-----|-----|-----|---|
| 1* | Primary or less | no | 15–39 years | Normal | no | no | 0.002 |
| 2 | Primary or less | **yes** | 15–39 years | Normal | no | no | 0.005 |
| 3 | Primary or less | no | 40–59 years | Normal | no | no | 0.124 |
| 4 | Primary or less | no | ≥60 years | Normal | no | no | 0.916 |
| 5 | Primary or less | no | 15–39 years | Overweight | no | no | 0.016 |
| 6 | Primary or less | no | 15–39 years | Obesity | no | no | 0.131 |
| 7 | Primary or less | no | 15–39 years | Normal | **yes** | no | 0.005 |
| 8 | Primary or less | no | 15–39 years | Normal | no | **yes** | 0.009 |
| 9 | Primary or less | **yes** | 40–59 years | Normal | no | no | 0.291 |
| 10 | Primary or less | **yes** | 40–59 years | Overweight | no | no | 0.789 |
| 11 | Primary or less | **yes** | ≥60 years | Overweight | no | no | 0.997 |
| 12 | Primary or less | **no** | ≥60 years | Overweight | **yes** | **yes** | 0.999 |
| 13 | Secondary | **yes** | 40–59 years | Obesity | **yes** | **yes** | 0.999 |
| 14 | Secondary | **yes** | ≥60 years | Obesity | **yes** | **yes** | 0.999 |
| 15 | University | **yes** | ≥60 years | Overweight | **yes** | **yes** | 0.999 |
| 16 | University | **yes** | ≥60 years | Obesity | **yes** | **yes** | ≈1,00 |

Ind.: individual, BMI: Body Mass Index, CKF: Chronic Kidney Failure, FHH: Familial hypertension

*Reference individual, p: probability of becoming hypertensive based on combination of risk factors.

over 60 years than young patients under 40 years. Patients with family history of hypertension had near three times higher (AOR: 2.9, 95% CI: 1.3–6.7, p: 0.007) the risk to become hypertensive than patients with non familial hypertension history. This risk was more than two times (AOR: 2.5, 95% CI: 1.1–5.7, p: 0.037) in overweight patients than normal and underweight patients. That risk of hypertension was more than 3 times higher (OR: 3.7, 95% CI: 1.2–11.5, p: 0.023) among obese patients than normal and underweight. Patients with chronic kidney failure had approximately 5 times (AOR: 4.9, 95% CI: 1.8–15.6, p: 0.003) the risk of hypertension than normal patients. The Wald test ($X^2$ = 86.7, df = 8, p<0.001) rejected the null hypothesis and therefore to confirm the alternative hypothesis stating that there is at least one estimate significantly different to zero. This suggests an overall significance of the model. Pearson residuals test of ($X^2$ = 266.17, df = 344, p = 0.99) was done and showed the model was well adjusted on the observations. A McFadden statistic ($R^2$: 0.4) also indicated that this model had a good fit.

The influential points' analysis based on Hoaglin and Welsh criterion showed that only 9 points were influential. Also, Cook's distance showed that only 3 points (108, 114 and 199) were outliers, which means the influential points were not numerous. Studentized residues analysis (Fig 1) showed that 97% (343/353) were between -2 and 2. Observations with residues (Fig 2) greater than 2 were ten (9, 34, 105, 108, 114, 199, 212, 232, 265, 273). Then, any observation with studentized residue were less than -2, indicating that the number of outliers were negligible.

## Cross validation and probabilities predictions

The Fig 3 showed the ROC curve and area under curve (AUC). A bootstrap method, with 2000 replications, was used to determine an AUC of 88.6% (95% CI: 84.1%-92.3%) which suggested an excellent discrimination (Fig 3). This implies that saturated model had an excellent predictive power and probabilities accurately determine patients with hypertension based on

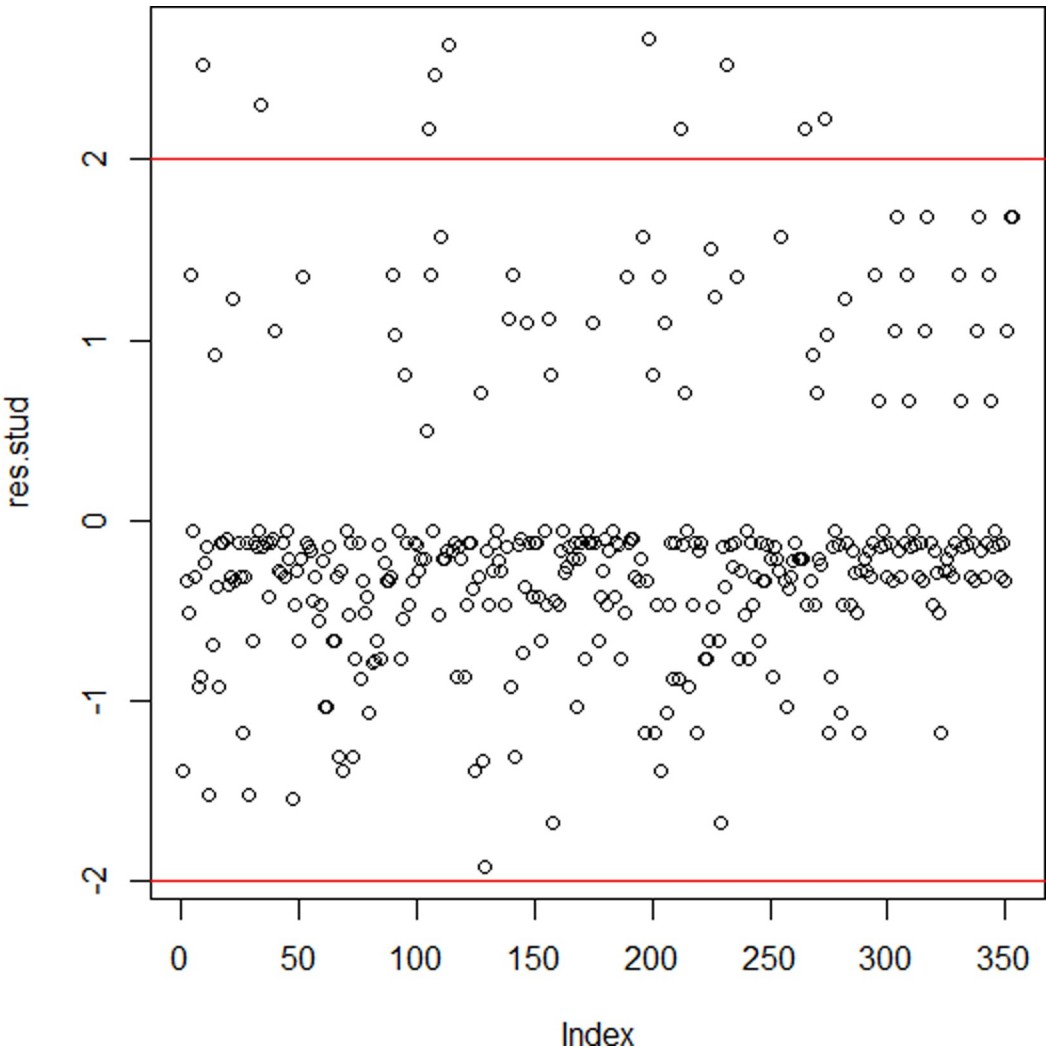

**Fig 1. Studentized residuals.**

identified characteristics. In supervised learning, the resubstitution error rate was 13.3%. This error rare is supposed to indicate the performance of the model when it is employed on the population. However, we know, being estimated on learning data, it was biased, and under estimating true error rate using cross-validation method which is the best quality estimator than the substitution method. The cross-validation error rate was 16.1%.

The table showed the predicted probabilities of becoming hypertensive based on different scenarios of having either a risk factor or a combination of risk factors. The first individual was the one without risk factors. He was considered as the reference individual. Sixteen predictions were generated from reference individual to whom with all hypertension risk factors. The table below showed increasing probabilities predictions from zero factors to all factors.

In absence of the risk factors highlighted above, the reference individual have 2 per 1000 the chance to become hypertensive. This probability goes from simple to more than double among smokers and 6 times higher among adult patients aged between 40 and 60 years. Then, no smokers patients with normal weight, aged between 40 and 59 years and the young obese patients (19–40 years) had probability to become hypertensive between 12% and 15%.

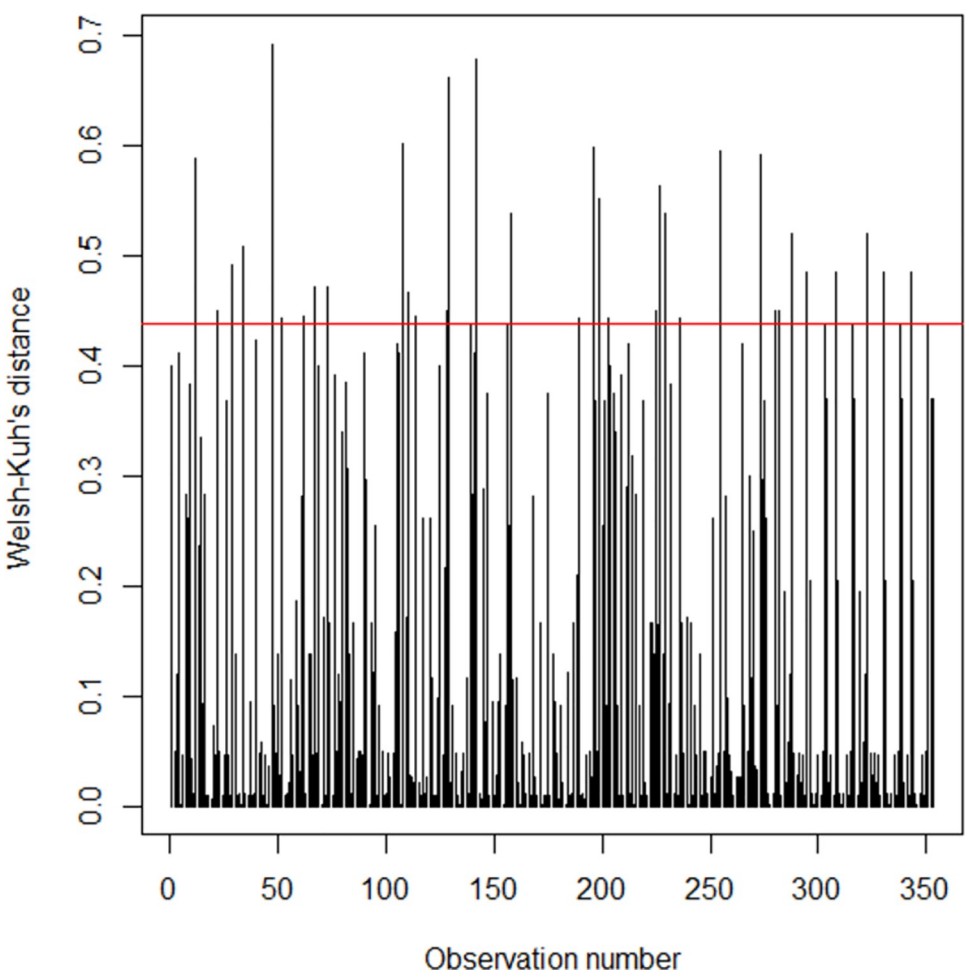

**Fig 2. Welsh-Kuh's distance.**

Comparatively to the reference patient, this probability was more than 40 times higher among patients aged between 60 years and over. If the individual aged between 40 and 60 was also smoker, his probability to become hypertensive increased near 15 times (29.1%). If that patient was overweight, that probability increased from 29.1% to 78.6%. Furthermore, if the overweight patients aged between 60 and over were also smoked, their chance to develop hypertension was 99.7%. That probability was very high among patients with coexistence of more than three risk factors (99%).

The highest probability was observed among patients who were at the same time with university education level smokers, chronic kidney failure and born in the hypertensive family. The 11th, 12th, 13th, 14th, 15th and 16th patient were respectively at 99.7%, 99.9%, 99.9%, 99.9% and 100% risk of hypertension (**Table 4**).

## Discussion

In this study, we determined the prevalence of hypertension, identified principal predictive risk factors of hypertension and predictive probabilities to become hypertensive based on risk factors. Overall, the prevalence of hypertension was relatively high Considering women only, the prevalence was also high. That prevalence was similar to results of a recent study conducted

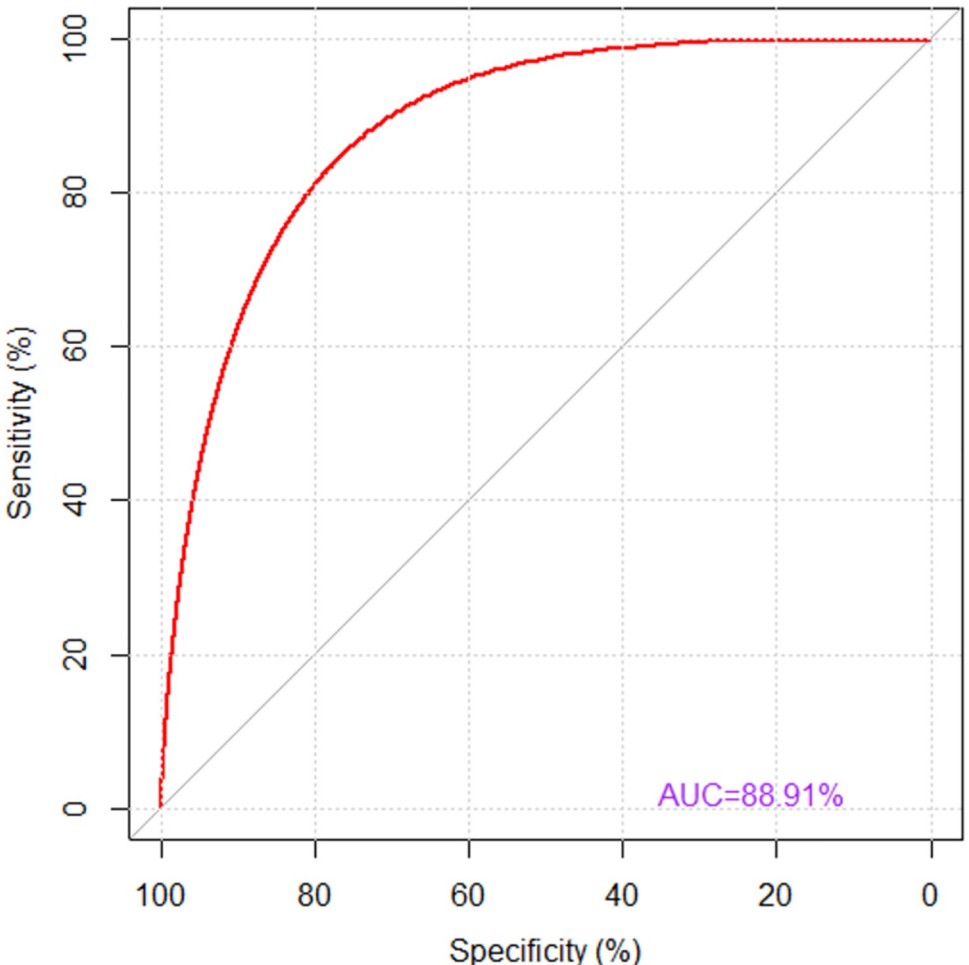

**Fig 3. Area under curve.**

in Lesotho which showed a relatively high prevalence (17.3%) among women [34]. These results were also in the line with previous study recently conducted in US among adults which showed a high prevalence among men than woman [35]. In another study conducted in Saudi Arabia, the prevalence was 15.2% among those aged 15 years old and above had different levels of hypertension [36].

This study did not showed the significant difference of the prevalence between women and men [36]. This finding was consistent with a studies conducted in Benin where there was no significant difference between men [37]. The highest prevalence of hypertension observed in diabetic patients and the lowest in young patients aged under 40 years [37]. This study showed that the hypertension's prevalence was less than SSA's hypertension prevalence. The pre hypertension was estimated on high proportion.

Literature on marital status and hypertension was inconclusive and mostly compared never married to currently married persons [38]. In congruence to this, our study did not show association between marital status and hypertension [38]. These findings were contrary to what had previously reported on this association [39]. After adjusting hypertension on other covariates using logistic regression model, high educational level, smoking, advanced age, overweight, obesity, chronic kidney failure and familial history of hypertension were significantly

associated with hypertension. Similar findings were found in a previous study conducted in Malawi, Kenya and Bangladesh which showed that that the factors associated with hypertension were overweight, smoking, education level and older age [40–42]. Similar findings were also reported in recent study conducted in Nepal which showed that being overweight, obese and with hypertension's family history were associated with hypertension [43]. The association between advanced age and high risk of hypertension could be due to the biological effects of increased arterial resistance which increases with old age [44]. This study didn't find the association between residence and marital status. Furthermore, as in this study, alcohol was also not associated with hypertension. These findings are in line with results reported in two studies conducted in Benin and some Europeans countries [34, 37].

This study showed that predicted probabilities to become hypertensive was low in young patients aged under 40 years. High probabilities were observed in patients with advanced age. It was also observed in patients with coexistence of risk factors. The highest probabilities (≥90%) were observed among old patients with at least two additional risk factors.

One strength of our study was the ability to study hypertensive and normotensive people at the same time, combining descriptive and inferential analysis (logistic regression with fixed effects, Wald test, deviance test) to build the ROC curve. Other strengths of the study were the ability to estimate area under curve, to build bootstrap AUC interval confidence using Bootstrap method with 2000 replicates, model's residuals analysis using Welsh-Kuh's distance, predicting probabilities of becoming hypertensive given a combination of risk factors. Lastly, even if the sample was not nationally representative to some extent, all patients were from all over the country towards these two hospitals (tertiary referral hospitals).

However, despite these strengths, some limitations should be noted during interpretation and policy formulation. First of all, this study was a cross-sectional study, which could only provide clues to the etiology, and further exploration was needed to prove the causal relationship; Secondly, our study used secondary data and as such, we were unable to measure quantities and type of alcohol and tobacco consumed as well as obtain information on physical activities which have been found to be associated with hypertension. Lastly, the sample size was relatively not large and caution should be taken when generalizing findings on high blood pressure as data used were only reported from two hospitals.

To validate findings, additional studies should be conducted in other hospitals in the country and take into others characteristics including more biomarkers. Score based models and nomograms should be used to support clinical implementation of risk models. A random effect logistic regression or Bayesian regression based on Markov chain Hamiltonian Monte Carlo simulations and Langevin algorithms could give precision in the estimation of model's parameters. Bayesian credibility intervals as such these methods are recommended for future research. The main interest of this study was to identify predictive risk factors of hypertension which allowed prediction of hypertension's occurrence controlling possible cofounders.

## Conclusion

This study showed that the hypertension prevalence was relatively high. Hypertension's prevalence was not significantly different in men and women. Predictive risk factors of hypertension were advanced age smoking, presence of chronic kidney failure, existing cardiovascular impairment, educational level and body mass index.

The lowest predicted probability of hypertension was observed in young patients with no risk factors. High predicted probabilities to become hypertensive were observed in patients with coexistence of two or more risk factors. Resources in Burundi are scare, therefore, the

tackling the high burden of cardiovascular diseases should be based on instituting systems for early detection and prompt treatment especially those identified as high risks.

To our knowledge, no study combining the predictive risk factors analysis and probabilities predictions have been carried in Burundi. At the community level, efforts should be channelled towards intensifying innovative and inclusive health promotion aimed at behaviour change. At the health system, creating a risk-based nomogram based on these identified risks factors could allow those at high risks to be identified early and well-targeted with the needed treatment.

Finally, provision of long term care for those identified cases will depend on not just consistent treatment but also on the overall health systems' strengthening. This will ensure sustainability and effectiveness of public health interventions aimed at chronic diseases tackling along with other high burden infectious diseases.

## Supporting information

**S1 Data.**
(SAV)

## Acknowledgments

The authors are thankful to all frontline workers and administration of both hospitals.

## Author Contributions

**Conceptualization:** Arnaud Iradukunda.

**Data curation:** Arnaud Iradukunda, Emmanuel Nene Odjidja, Stephane Karl Ndayishima, Cheilla Izere, Nestor Ntakaburimvo.

**Formal analysis:** Arnaud Iradukunda, Emmanuel Nene Odjidja, Cheilla Izere, Nestor Ntakaburimvo.

**Investigation:** Arnaud Iradukunda, Emmanuel Nene Odjidja.

**Methodology:** Arnaud Iradukunda, Emmanuel Nene Odjidja, Stephane Karl Ndayishima, Cheilla Izere.

**Software:** Egide Ngendakumana, Gabin Pacifique Ndayishimiye, Darlene Sinarinzi, Cheilla Izere, Nestor Ntakaburimvo.

**Supervision:** Emmanuel Nene Odjidja, Egide Ngendakumana, Arlene Akimana.

**Visualization:** Arnaud Iradukunda, Emmanuel Nene Odjidja.

**Writing – original draft:** Arnaud Iradukunda, Emmanuel Nene Odjidja, Egide Ngendakumana, Gabin Pacifique Ndayishimiye, Cheilla Izere, Arlene Akimana.

**Writing – review & editing:** Arnaud Iradukunda, Stephane Karl Ndayishima, Egide Ngendakumana, Gabin Pacifique Ndayishimiye, Darlene Sinarinzi.

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
