## [Decision Letter · Decision Letter 0]

23 Jul 2021

PONE-D-21-17418

Prevalence and Predictive Risk Factors of Hypertension in patients hospitalized in Kamenge Military hospital and Kamenge University teaching hospital in 2019: A Fixed Effect Modelling Study in Burundi

PLOS ONE

Dear Dr. Iradukunda,

Thank you for submitting your manuscript to PLOS ONE. After careful consideration, we feel that it has merit but does not fully meet PLOS ONE’s publication criteria as it currently stands. Therefore, we invite you to submit a revised version of the manuscript that addresses the points raised during the review process.

Please revise the manuscript in line with Reviewers' comments, and particularly address Reviewers' comments regarding the dataset, data and performed analyses in the revised manuscript.

We look forward to receiving your revised manuscript.

Kind regards,

Irena Ilic, M.D., Ph.D

Academic Editor

PLOS ONE

Journal Requirements:

Reviewers' comments:

Reviewer's Responses to Questions

**Comments to the Author**

1. Is the manuscript technically sound, and do the data support the conclusions?

Reviewer #1: Yes

Reviewer #2: Partly

2. Has the statistical analysis been performed appropriately and rigorously? 

Reviewer #1: Yes

Reviewer #2: N/A

3. Have the authors made all data underlying the findings in their manuscript fully available?

Reviewer #1: Yes

Reviewer #2: Yes

4. Is the manuscript presented in an intelligible fashion and written in standard English?

Reviewer #1: Yes

Reviewer #2: No

5. Review Comments to the Author

Reviewer #1: The manuscript about Prevalence and Predictive Risk Factors of Hypertension in patients hospitalized in Kamenge Military hospital and Kamenge University teaching hospital is well written with detailed descriptions of methodology and results. Numerous advanced statistical methods have been applied. The Table and Figures are quite detailed. Both the topic of the study as well as the findings are interesting. However, I have some comments about the manuscript, as described below.

1. The attached database has 304 respondents, while the manuscript shows 353. It is not possible to repeat the analyzes listed in the paper. A proper database needs to be attached.

2. The population of respondents is not precisely explained through the structure of wards and patients in hospitals. What are the primary diagnoses for which they were admitted to the hospitals? How was the randomization done in relation to the hospital ward?

3. When calculating the sample size, the prevalence of hypertension was taken to be 0.5, which is a large value.

4. The incidence of Chronic kidney failure is 52.1% (184/353 * 100), which is a high incidence. Additionally, whether Chronic kidney failure is a predictor or a consequence of hypertension. Can these patients be regarded as predictors of hypertension in the study?

5. It is not explained how and based on what data was the General cardiovascular risk calculated and what it refers to.

6. More than 50% of references are older than 5 years. There are newer publications for this topic, so I suggest adding them.

7. How the results obtained on the basis of hospital patients can be generalized to the population of Burundi?

8. Include BMI in the model as an ordinal variable with all categories or compare Overweight and Obesity in relation to Underweight and Normal.

9. Is cardiovascular comorbidity a predictor or a consequence of hypertension?

10. In logistic regression, the odds ratio is obtained, so the results should be interpreted in that way.

11. The paper does not state the level of significance.

12. Delete AIC values from tables with univariate analyzes because they do not contribute to informativeness.

13. In the attached database it is necessary to check the values for: age (range 15-107 years) and BMI (minimum value is 6.72)

14. In Table 3 the p-value for Smoking is 0.081, which is not statistically significant.

15. In Table 1 the sum of all categories in column N is 353. In cell named Total, the value is 309.

16. All numeric values in tables and text should be set to one decimal place.

I suggest a revision of the manuscript followed by second round of review.

Reviewer #2: The topic of the paper is interesting, always actual and always need new information-up to date. The topic is also relevant in the application of appropriate statistical modeling to identify risk factors.

In order for the paper to be publish, it is necessary to change the way of describing the used statistical methods and presenting the results. In the material and methods, it is described how the required sample size was calculated. It only needs to be describe, without writing formulas.

In the part of statistical analysis, it should be described why certain statistical tests and analyzes were used. Describe the tests used in the different data collected and analyzed. It is not necessary to show the formulas of the regression analysis used. This is unnecessary if the topic of the paper is hypertension in a certain patient population. This way of writing the methodology as well as presenting the results is matching with the paper whose topic is logistic regression analysis, with example of a study to identify risk factors for the occurrence of hypertension in the observed patient population.

The results are needs to be completely changed. Show the tables so that the results are clear to the doctor who reading the paper.

Table 1: What 95% CI refers. Table 2: explain what AIC is. Is table 3 required?

Fig. 1 and Fig. 2, and Fig. 4 are unclear. Describe the adequacy of the model, and the tests used for testing in the methodology. I don’t think they need to be shown in the results.

6. PLOS authors have the option to publish the peer review history of their article (what does this mean?). If published, this will include your full peer review and any attached files.

Reviewer #1: No

Reviewer #2: No

---

## [Author Response · Author response to Decision Letter 0]

21 Sep 2021

Reviewer 1

Comment 1: The attached database has 304 respondents, while the manuscript shows 353. It is not possible to repeat the analyzes listed in the paper. A proper database needs to be attached.

Response 1: Thank you for the comment. A dataset of 353 respondents have been attached. 

Comment 2: The population of respondents is not precisely explained through the structure of wards and patients in hospitals. What are the primary diagnoses for which they were admitted to the hospitals? How was the randomization done in relation to the hospital ward? 

Response 2: Thank you for the comments. Description have been made in the manuscript

Comment 3: When calculating the sample size, the prevalence of hypertension was taken to be 0.5, which is a large value.

Response 3: Yet, studies to understand the epidemiology and associated in the context of Burundi are lacking. Without prevalence of hypertension in previous studies conducted in Burundi, we used p=0.5 as recommended in literature on sampling method (Giezendanner, François Daniel. "Taille d’un échantillon aléatoire et marge d’erreur." Instruction Publique, Culture et Sport (2012): 7.) . 

Comment 4: The incidence of Chronic kidney failure is 52.1% (184/353 * 100), which is a high incidence. Additionally, whether chronic kidney failure is a predictor or a consequence of hypertension. Can these patients be regarded as predictors of hypertension in the study?

Response 4: This incidence is relatively high because the study included all patients from intensive care unit with chronic kidney failure. Majority of these patients are hospitalized there because this areas are located together with haemodialysis unit. Therefore, these patients can be regarded as predictors as the Renin-Angiotensin-Aldosterone System participate in the development of hypertension from chronic kidney failure. 

Comment 5: It is not explained how and based on what data was the General cardiovascular risk calculated and what it refers to.

Response 5: The cardiovascular risk was based on both blood pressure level, risk factors and clinical symptoms. The increase of blood pressure level associated with increase of risk factors (them founded among our patients) increase the risk. In the data set, we scored each risk factors by one 1 point. For each patients, calculated the total of the points and we classified it in four classes: Zero risk factor,1-2 Risk factors, 3 and more risk factors and 3 risk factors associated with clinical blood pressure.

Therefore, as this evaluation is not linked with title and specific objectives of the study, the table 5 have been removed in the manuscript.

Comment 6: More than 50% of references are older than 5 years. There are newer publications for this topic, so I suggest adding them.

Response 6: New publications are added. 

Comment 7: How the results obtained on the basis of hospital patients can be generalized to the population of Burundi?

Response 7: Partially, we generalized these findings because these hospitals are the tertiary national reference hospitals which receive patients from across the country. On other side, these findings could not be generalized because the sample is not nationally representative. We highlighted it is as limit of the study, prompt to conduct others study on the national level. In conclusion, these findings cannot be generalized and suggest to conduct another study in most of Burundian hospitals. 

Comment 8: Include BMI in the model as an ordinal variable with all categories or compare Overweight and Obesity in relation to Underweight and Normal.

Response 8: BMI have been included in the model. It has been categorized as suggested 

Comment 9: Is cardiovascular comorbidity a predictor or a consequence of hypertension?

Response 9: By mention of cardiovascular comorbidities, we implied the presence of an existing cardiovascular impairment apart from hypertension. We have now clarified that throughout the manuscript. Thank you. 

Comment 10: In logistic regression, the odds ratio is obtained, so the results should be interpreted in that way.

Response 10: Results have been presented using odds ratios and adjusted odds across the manuscript. Interpretation follows same 

Comment 11: The paper does not state the level of significance.

Response 11: Level of significance has been stated on page 6 line 7.

Comment 12: Delete AIC values from tables with univariate analyzes because they do not contribute to informativeness.

Response 12: Changes have been made in the manuscript 

Comment 13: In the attached database it is necessary to check the values for: age (range 15-107 years) and BMI (minimum value is 6.72)

Response 13: These values are correct even if they seems to outliers. 

Comment 14: In Table 3 the p-value for Smoking is 0.081, which is not statistically significant.

Response 14: Even if the table 3 have been removed in manuscript, the selection of significant variables was based on AIC and the best model is the one with low AIC. As we used the backward method, when we remove tobacco consumption, the AIC increases which is not recommended in modelling. Beside the Akaike criterion, the model without tobacco decrease the AUC, and consequently the predictive power. All these augments allowed us to include tobacco use in the model. 

Comment 15: In Table 1 the sum of all categories in column N is 353. In cell named Total, the value is 309.

Response15: Changes have been made in the manuscript

Comment 16: All numeric values in tables and text should be set to one decimal place.

Response 16: Changes have been made in the manuscript 

Reviewer 2

The topic of the paper is interesting, always actual and always need new information-up to date. The topic is also relevant in the application of appropriate statistical modelling to identify risk factors.

Comment 1: In order for the paper to be publish, it is necessary to change the way of describing the used statistical methods and presenting the results. In the material and methods, it is described how the required sample size was calculated. It only needs to be describe, without writing formulas.

Response 1: Thank you for the comment. All those changes have been made in the manuscript. 

Comment 2: In the part of statistical analysis, it should be described why certain statistical tests and analyzes were used. Describe the tests used in the different data collected and analyzed

Response 2: Changes have been made in the manuscript 

Comment 3: It is not necessary to show the formulas of the regression analysis used. This is unnecessary if the topic of the paper is hypertension in a certain patient population.

Response 3: Changes have been made in the manuscript

Comment 4: This way of writing the methodology as well as presenting the results is matching with the paper whose topic is logistic regression analysis, with example of a study to identify risk factors for the occurrence of hypertension in the observed patient population.

Response 4: changes have been made in the manuscript

Comment 5: Table 1: What 95% CI refers. 

Response 5: 95% CI refers to proportion hypertension ‘’P+ ‘’

Comment 6: Table 2: explain what AIC is.

Response 6 : AIC have been removed in the Table 3 but is very well explained in manuscript as Akaike Information Criterion used to choose best model.

Comment 7: Is table 3 required?

Response 7: No it is not required and we removed it in the manuscript 

Comment 8: Fig. 1 and Fig. 2, and Fig. 4 are unclear. Describe the adequacy of the model, and the tests used for testing in the methodology.

Response 8: Changes have been made in the manuscript.

---

## [Decision Letter · Decision Letter 1]

19 Oct 2021

PONE-D-21-17418R1Prevalence and Predictive Risk Factors of Hypertension in patients hospitalized in Kamenge Military hospital and Kamenge University teaching hospital in 2019: A Fixed Effect Modelling Study in BurundiPLOS ONE

Dear Dr. Iradukunda,

Thank you for submitting your manuscript to PLOS ONE. After careful consideration, we feel that it has merit but does not fully meet PLOS ONE’s publication criteria as it currently stands. Therefore, we invite you to submit a revised version of the manuscript that addresses the points raised during the review process. Thank you to authors for addressing the comments. Please, carefully review and answer the comments raised by a reviewer.

We look forward to receiving your revised manuscript.

Kind regards,

Irena Ilic, M.D., Ph.D

Academic Editor

PLOS ONE

Journal Requirements:

Additional Editor Comments (if provided):

Reviewers' comments:

Reviewer's Responses to Questions

**Comments to the Author**

1. If the authors have adequately addressed your comments raised in a previous round of review and you feel that this manuscript is now acceptable for publication, you may indicate that here to bypass the “Comments to the Author” section, enter your conflict of interest statement in the “Confidential to Editor” section, and submit your "Accept" recommendation.

Reviewer #1: (No Response)

Reviewer #2: All comments have been addressed

2. Is the manuscript technically sound, and do the data support the conclusions?

Reviewer #1: Yes

Reviewer #2: Yes

3. Has the statistical analysis been performed appropriately and rigorously? 

Reviewer #1: Yes

Reviewer #2: Yes

4. Have the authors made all data underlying the findings in their manuscript fully available?

Reviewer #1: Yes

Reviewer #2: Yes

5. Is the manuscript presented in an intelligible fashion and written in standard English?

Reviewer #1: Yes

Reviewer #2: Yes

6. Review Comments to the Author

Reviewer #1: I suggest that the manuscript can be accepted for publication after a minor revision:

1. The attached database does not have a variable BMI divided to categories (Underweight and healthy, Overweight, Obese)

2. Check all p-values (for example, the p-value for EL in Table 2 is different from the p-value obtained after analysis from the database)

3. Check and correct all errors in the Tables (one of the examples [2.1-12.4-11.15])

4. Equalize decimal places for numbers – number of decimal places should be consistent throughout the paper

5. Correct typing mistakes, sometimes there is no break, and sometimes it is in the wrong place.

Reviewer #2: (No Response)

7. PLOS authors have the option to publish the peer review history of their article (what does this mean?). If published, this will include your full peer review and any attached files.

Reviewer #1: No

Reviewer #2: No

---

## [Author Response · Author response to Decision Letter 1]

28 Oct 2021

Journal Requirements: 

Response: We confirm the reference list is complete and accurately reflects all the in-text citations. There are no cited papers which have been retracted and the list as it stands need no further changes. Thank you.

Reviewer comments and their respective responses 

Reviewer #1: I suggest that the manuscript can be accepted for publication after a minor revision:

Comment 1. The attached database does not have a variable BMI divided to categories (Underweight and healthy, Overweight, Obese)

Response: Thank you for the comment. As we used R software, we coded directly the BMI’s categories in R: These are the code used to carry out the BMI and its categories: 

a. BMI<-(WEIGHT/ (HEIGHT^2)) #Body mass index calculation

b. BMI cat<- ifelse (BMI<25,"Underweight/Normal", (BMI<30,"Overweight", "Obesity")) #Body mass index categorization.

Coment.2. Check all p-values (for example, the p-value for EL in Table 2 is different from the p-value obtained after analysis from the database)

Response 2: Thank you for your comments. All p value were have been well checked. 

Comment 3. Check and correct all errors in the Tables (one of the examples [2.1-12.4-11.15])

Response 3: Thank you for the comment. Change have been made in the main manuscript 

Comment. 4. Equalize decimal places for numbers – number of decimal places should be consistent throughout the paper

Response 4: Changes are made in the main manuscript

Comment. 5. Correct typing mistakes, sometimes there is no break, and sometimes it is in the wrong place.

Response 5: Changes are made in the main manuscript

Reviewer #2: (No Response)

Kind regards,

Arnaud IRADUKUNDA (Corresponding author)

---

## [Editor Report · Decision Letter 2]

5 Nov 2021

Prevalence and Predictive Risk Factors of Hypertension in patients hospitalized in Kamenge Military hospital and Kamenge University teaching hospital in 2019: A Fixed Effect Modelling Study in Burundi

PONE-D-21-17418R2

Dear Dr. Iradukunda,

We’re pleased to inform you that your manuscript has been judged scientifically suitable for publication and will be formally accepted for publication once it meets all outstanding technical requirements.

Kind regards,

Irena Ilic, M.D., Ph.D

Academic Editor

PLOS ONE
---

## [Editor Report · Acceptance letter]

22 Nov 2021

PONE-D-21-17418R2 

Prevalence and Predictive Risk Factors of Hypertension in patients hospitalized in Kamenge Military hospital and Kamenge University teaching hospital in 2019  : A Fixed Effect Modelling Study in Burundi 

Dear Dr. Iradukunda:

I'm pleased to inform you that your manuscript has been deemed suitable for publication in PLOS ONE. Congratulations! Your manuscript is now with our production department. 

Kind regards, 

on behalf of

Dr. Irena Ilic 

Academic Editor

PLOS ONE